# Music as Add-On Therapy in the Rehabilitation Program of Parkinson’s Disease Patients—A Romanian Pilot Study

**DOI:** 10.3390/brainsci11050569

**Published:** 2021-04-29

**Authors:** Dana Marieta Fodor, Xenia-Melania Breda, Dan Valean, Monica Mihaela Marta, Lacramioara Perju-Dumbrava

**Affiliations:** 1Department of Neuroscience, “Iuliu Hatieganu” University of Medicine and Pharmacy, 400012 Cluj-Napoca, Romania; breda_xenia@yahoo.co.uk (X.-M.B.); lperjud@gmail.com (L.P.-D.); 2Regional Institute of Gastroenterology and Hepatology, 400162 Cluj-Napoca, Romania; valean.d92@gmail.com; 3Department of Medical Education, “Iuliu Hatieganu” University of Medicine and Pharmacy, 400202 Cluj-Napoca, Romania; mmarta@umfcluj.ro

**Keywords:** Parkinson’s disease, music therapy, neurorehabilitation, quality of life

## Abstract

Music has been proven to have therapeutic potential in neurological disorders, especially Parkinson’s disease (PD), since rhythmic auditory cueing can partially replace the progressive loss of rhythmicity and automaticity. Several reports have highlighted improvements in motor outcomes in PD patients undergoing music therapy, but only a few studies have evaluated non-motor outcomes, such as quality of life (QoL), which deteriorates with disease progression. The current pilot study aims to examine the effects of a multimodal rehabilitation program centered on physical therapy combined with listening to music on self-reported QoL in people with PD, compared to the same rehabilitation program alone. The study was conducted on patients with idiopathic PD who attended a specific rehabilitation program with a duration of 2.5 h daily for 14 days. The patients were divided into the study group (16 patients), who listened to background music during the rehabilitation program sessions, and the control group who did not listen to music during sessions. The patients were assessed using the self-report Parkinson’s Disease Questionnaire (PDQ-39) at the beginning of the program and 1 month after its initiation. The patients in the study group registered greater improvements in five of the eight areas of life assessed by PDQ-39 compared to the control group. In conclusion, listening to music combined with a multimodal rehabilitation program centered on physical therapy may be beneficial for the patients’ quality of life.

## 1. Introduction

Music represents a special stimulus, with a strong and complex modulatory effect on the central nervous system, as well as a demonstrated emotional, cognitive and motor impact. Music is capable of stimulating neuroplasticity both functionally and structurally through the activation of several interrelated neural networks, thus having therapeutic value in neurological diseases such as stroke, multiple sclerosis, traumatic brain injury and Parkinson’s disease [1,2,3,4].

In PD, the patient is confronted with an alteration of rhythmicity and automaticity due to a progressive decrease of dopamine levels in the corticostriatal network, which results in the progressive alteration of self-initiated movements [1]. Studies have shown that rhythmic entrainment through rhythmic auditory cueing, alone or through music therapy, can provide external cues that facilitate the initiation and synchronization of movements. This is because it opens new pathways that enhance auditory–motor coupling at the cortical level and activates the cerebellum–thalamic–cortical circuitry which bypasses the impaired striatal-cortical circuitry [2,3,4,5,6,7,8]. Thus, these patients see an improvement in walking ability with respect to kinematic parameters (such as velocity, step cadence, step amplitude and stride length [9,10]), balance and freezing phenomena [11], as well as bradykinesia [12] and tremors [13] as a result of better perceptual and motor timing [14]. Improvements in non-motor symptoms such as anxiety, depression or cognitive functions have also been described as effects of music therapy [12,15,16].

The question is whether rhythmic auditory stimulation cueing alone, through a metronome, or its inclusion in the music to which the patient is exposed is better at helping patients with Parkinson’s disease [4,17]. A partial answer is given by the conclusions of some studies showing that listening to pleasurable music produces a highly emotional response correlated with the induction of dopamine release in the ancient circuitry of the mesolimbic reward system [18,19]. However, listening to music has an intrinsic rewarding value involving two phases, ‘wanting’ and ‘liking’, each of them activating different anatomical dopaminergic circuits. The anticipatory phase is related to following the temporal unfolding of the musical stimulus, including cues that can trigger the expectation of pleasure from the desired musical sequences (‘wanting phase’) and is linked to the activation of the dorsal striatum (substantia nigra, ventral tegmentum, caudate). The second phase is represented by the peak of the emotional response, consisting of the fulfillment of the expectation and the confirmation of the reward prediction (‘liking phase’) and is related to the dopaminergic activation of the ventral striatum, especially by the nucleus accumbens, which in turn is connected to the hypothalamus and insula (suggesting a regulatory autonomic and physiological response effect as well) and to the orbitofrontal cortex (involved in cognitive processing and decision-making) [18,19,20,21]. In conclusion, rhythmic music therapy and to an even greater extent, music-based movement therapy brings additional benefits through the emotional and cognitive modulation effect [22].

In addition to all these objective measurements concerning motor, emotional and cognitive improvements caused by rhythmic music therapy, the way in which PD patients perceive its impact on their quality of life (QoL) is also of high interest. This is an important aspect given that PD patients report a deterioration in their daily life with the progression of the disease. The 39-item Parkinson’s Disease Questionnaire (PDQ-39) is the most thoroughly validated self-report measure used to assess health-related QoL in PD patients. PDQ-39 is a multidimensional scale which evaluates the impact of PD on 8 areas of life (mobility, activities of daily living (ADLs), emotional well-being, stigma, social support, cognition, communication and bodily discomfort), quantifying how often the patient experiences difficulties in each of them [23]. Paccheti et al. (2000) found a considerable improvement in QoL in the music therapy group after 13 weekly sessions of instrumental musical improvisation compared to physiotherapy, but this finding was not sustained 2 months after therapy [12]. Similarly, Craig et al. (2006) reported a beneficial effect on QoL in the music therapy group compared to the neuromuscular therapy group, sustained at 8 days after receiving treatment twice a week for 4 weeks [13]. Pohl et al. (2020) showed that a 12-week Ronnie Gardiner Rhythm and Music (RGRM) program can induce an improvement in QoL [15]. Spina et al. (2016) found an increase in QoL for the global PDQ-39 score, especially for the emotional well-being domain, after 24 weeks of weekly treatment, an effect that was maintained for 6 months compared to controls, who experienced a deterioration in the mobility and bodily discomfort domains during this time [16].

The effects of music therapy in PD patients can undoubtedly vary depending on preferences for certain music genres and the degree of acquaintance with that music. As discussed above, it is enjoyable music that activates the reward-motivational circuitry [18,19,20,21]. The enjoyment of music can be considerably increased by familiarity with the music. Studies have shown that exposure to music with which the patient is acquainted allows better tempo matching with rhythmic auditory stimulation because a familiar beat structure needs less cognitive control for synchronization [24]. Familiar music favors prediction and anticipation [18]. Moreover, preferred familiar music gives greater enjoyment and increases emotional engagement as well as compliance with the rehabilitation training programs [25,26].

Classical music improves the spatiotemporal parameters of gait by simultaneously reducing gait speed and increasing stride duration, while also decreasing anteroposterior trunk tilt range of motion, which may ameliorate festination. Conversely, rock and heavy metal music increase pelvis obliquity and rotation range of motion, suggesting an amelioration of rigidity [27].

We hypothesized that predictable rhythmic music associated with a multimodal rehabilitation program, centered on physical therapy, could improve PD patients’ quality of life because (i) it improves motor symptoms through rhythmic auditory cueing, and (ii) it is a demonstrated emotional modulator, which could impact on the anhedonia and depression occurring in PD. Consequently, this pilot study aims to assess this hypothesis as well as the feasibility of such a therapeutic approach.

## 2. Materials and Methods

This study was a randomized, single-blinded quality of life pilot study.

### 2.1. Patients

The participants included in the study were recruited from PD patients who applied for a known rehabilitation program offered on a voluntary basis within the ‘Power of hope for Parkinson’s disease patients’ project organized in Cluj-Napoca, Romania. This monthly program offers classes of 10 patients each (which is the maximum capacity allowed by human and material resources), who attend daily 2.5 h sessions for 2 weeks (excepting Sundays), during which they undergo a program of physical therapy, art therapy and relaxation techniques alternating with breaks for rest and socialization. The study was initially intended to include 40 patients (4 classes of 10 patients each), randomly assigned: 2 classes (20 patients) as the study group and 2 classes (20 patients) as the control group. Of the 40 patients, 2 refused to participate and asked to be reassigned to another class of patients that was not included in the study. All of the remaining 38 patients gave their written informed consent to participate.

The inclusion criteria were a diagnosis of idiopathic PD and a stable dopaminergic treatment scheme in the last month and throughout the study (levodopa, levodopa plus dopamine agonist and/or MAO-B inhibitors). Atypical and secondary Parkinsonism, hearing loss, marked cognitive deficits and Hoehn and Yahr stages >III were exclusion criteria. After screening, 6 patients were excluded from the study (2 patients with atypical or secondary Parkinsonism, 1 patient with insufficiently corrected hearing impairment, 3 patients with modifications of the therapeutic scheme in the last month).

The 32 patients could choose to participate in one of the 4 classes depending on their preference/availability for a certain period, and the 4 classes (of 8 patients each) were randomly assigned: 2 classes (16 patients) as the study group and 2 classes (16 patients) as the control group. The person who scheduled the patients was blinded to the result of randomization. Ineligible patients who wanted to attend the program could join the classes but were not taken into consideration for analysis.

### 2.2. Procedure

The study group (consisting of 16 patients, 12 females and 4 males, aged between 60 and 76 years, mean age = 67 years, Hoehn and Yahr Scale stage I–III) was exposed during the 2.5 h sessions to classical music, played in the background. The patients were encouraged, whenever possible, to synchronize their walking and other movements with the rhythm of music. The control group (consisting of 16 patients, 10 females and 6 males aged between 56 and 74 years, mean age = 65 years, Hoehn and Yahr Scale stage I–III) attended the same rehabilitation program without music exposure. The characteristics of the group are presented in Table 1.

### 2.3. Interventional Program

The rehabilitation program was multimodal. Each session was attended by both groups and was structured similarly with a focus on physical therapy, but it also included an art therapy component and relaxation techniques, with breaks for rest and socialization. The objectives were to correct posture and gait, to reduce bradykinesia and rigidity and to improve upper limb dexterity using standing and walking balance exercises, stretching and joint flexibility exercises, muscle strengthening training, cardiovascular training and art therapy. The exercises were individualized when needed and slowly increased in intensity and complexity during the program. The details of the program are included in Table 2.

On screening, the patients completed a questionnaire related to their preferences for 3 music genres as well as to the music genre they would like to listen to during the rehabilitation program: classical, pop or rock music. The music preferences were divided among 50% (16) pop music, 25% (8) classical music and 25% (8) rock music. However, as far as music during the rehabilitation program was concerned, the answers were as follows: classical music was favored by 44% (14), pop music by 31% (10) and rock music by 25% (8); 6 patients who generally preferred pop music chose to listen to classical music during the rehabilitation training period. Mozart, Beethoven and Vivaldi were the most frequently preferred classical music composers by the study participants.

After completion of the rehabilitation program, the study group patients were instructed to continue listening to the same classical music for 2.5 h daily over the next 2 weeks, alongside their daily activities. According to the patients’ reports, the degree of task accomplishment ranged between 50% and 100%.

### 2.4. Outcome Assessment

Each patient completed the PDQ-39 questionnaire at the beginning of the rehabilitation program and 1 month after its initiation (2 weeks after its completion). The assessor was blinded to group allocation; however, it was not possible to blind the patients and the practitioners who worked with the patients.

### 2.5. Statistical Analysis

Microsoft Excel 2019 was used to create the database. IBM Microsoft SPSS v20.0 software was employed to interpret the data. The T test for independent samples was used to compare the mean values (taking into consideration the mean and standard deviation). The median values were compared using the Mann–Whitney U test, taking into account the mean value and 25–75 percentiles. A threshold value of *p* < 0.05 was considered statistically significant.

## 3. Results

By comparing the mean values of the results of the PDQ-39 scale (using the T-test for independent samples adjusted for multiple comparisons, i.e., the datasets conforming to a normal data distribution), statistically significant results were obtained between the final and initial evaluation in both groups. Regarding the eight areas of PDQ-39, the study group showed statistically significant differences in all eight, while in the control group, statistically significant differences were observed in four areas (mobility, emotional well-being, stigma and bodily discomfort) (Table 3 and Table 4).

By further comparing the median values of differences between the initial and final values between the two groups (using the Mann–Whitney U test, the distribution of datasets not being normal), greater statistically significant improvements were detected in the study group compared to the control group, both for the global PDQ-39 scale value (*p* < 0.001) and for the areas of ADLs (*p* = 0.002), emotional well-being (*p* = 0.001), social support (*p* = 0.02), communication (*p* = 0.002) and bodily discomfort (*p* = 0.002). (Table 5).

## 4. Discussion

The association of rhythmic music with a multimodal rehabilitation program centered on physical therapy in PD patients enhanced the effects of rehabilitation therapy on QoL, reflected in the self-report PDQ-39 scale. Compared to the control group without music in the rehabilitation program, the addition of music statistically significantly improved the global PDQ-39 scores, in accordance with previous reports regarding music therapy in general [12,13,16]. Concerning the eight life areas of PDQ-39, we found greater improvements in all of them compared to the control group, which were statistically significant in five areas: ADLs, emotional well-being, social support, communication and bodily discomfort. As far as the effect of music therapy on QoL in PD patients is concerned, PDQ-39 score has been mainly analyzed as a global value rather than in relation to the eight areas of life. A significant improvement in emotional well-being through active music therapy over 24 weeks was reported by Spina et al. [16]. In our case, the improvement in emotional status may have also modulated the perception of non-motor symptoms (bodily discomfort) and have had a favorable impact on the degree of functionality reflected by ADLs. An improvement in social support was also noticed in the group exposed to music, which suggests a possible alteration of the subjective perception of this aspect, possibly also due to the emotional modulatory effect of music. PD patients can progressively develop hypokinetic dysarthria. The improvement of communication in the study group, including an amelioration of dysarthria, is encouraging.

In addition, being intrinsically motivating, music could increase the efficacy of the rehabilitation program with which it is associated, compared to the less rewarding conventional physical therapy, because it ensures increased compliance with treatment.

Concerning the choice of classical music by the majority of the patients in the study, this seems to be a beneficial one because it has a melodic architecture of optimal complexity between predictability and the capacity to surprise [28]. We are aware that music preferences can vary among PD patients, like in the case of the general population, but we relied on the options expressed by the majority of patients participating in the study and on the relatively high degree of acquaintance with classical music in the urban population of Cluj-Napoca.

We decided to focus on QoL as an outcome using PDQ-39, even if this is a self-report scale whose results are potentially influenced by the patients’ subjectivity, for two reasons: (i) compared to a clinical evaluation that would involve a higher degree of objectivity but might reflect the patients’ motor or mood fluctuations, we considered that PDQ-39 better reflects the changes occurring in the patients’ status, being a summation of these fluctuations over the last month, and (ii) despite the motor improvement through pharmacological and non-pharmacological treatments, most PD patients report a progressive degradation of their daily QoL, so anything that might positively influence it would be beneficial [29].

There are several limitations to this pilot study. One is the small number of patients and the short duration of the study, which did not allow us to assess the duration of these effects. The difference between our study and other literature studies related to music therapy as an interventional program was that we ran an intensive 2-week program, whereas in other studies, the sessions were conducted 1–2 times a week over the duration of 4 to 12 weeks (maximum 24 in one study) [12,13,15,16]. Compared to other studies, our rehabilitation program was multimodal, including not only physical therapy, even if this was the main activity, but also some elements of art therapy and socialization, which could have contributed to the improvements of QoL in both groups. Another aspect is the contribution of the further two weeks of listening to the same classical music to the improvement in QoL observed in the study group. This raises the question of whether listening to music outside of rehabilitation sessions, including music alongside daily activities, could help to maintain the effects on QoL. Another limitation is given by the preferences for different music genres and the choice of classical music which was preferred by most but not all patients. Some of the patients, for whom classical music was not among their musical preferences or who were not acquainted with it, might have had a better emotional responses and engagement if they had listened to their favorite music, since it has been observed that preferred music in particular induces the activation of the dopaminergic mesolimbic circuit [17,18,22]. Assessing the effect of music on the QoL of PD patients based only on their listening to it and trying to synchronize movements with its rhythm (whenever possible) constitutes another study limitation. It would be interesting to explore other dimensions of music, for example, singing or playing an instrument (active music therapy, which is more engaging).

In Romania, such an approach combining rehabilitation therapy with listening to music, in small classes for 2–3 h daily over 2 weeks (combining physical therapy with art-therapy and socialization), is more feasible in urban areas (due to shorter distances to be covered by PD patients to attend such a class) and for retired persons with PD (who have more time available). For other PD patients, a rehabilitation program involving 1–2 meetings per week would be more suitable and might extend over a longer period. Listening to music at home between the sessions of the rehabilitation program or after completing it could be adapted to patients listening to the music of their choice, not necessarily to the same music used in the rehabilitation sessions, to improve engagement. Otherwise, music is almost a free resource, easy to apply as intervention and relatively widely accepted.

## 5. Conclusions

Combining music with a multimodal rehabilitation program centered on physical therapy had a beneficial effect on the quality of life of PD patients, reflected on global PDQ-39, especially on five of its life areas: ADLs, emotional well-being, social support, communication and bodily discomfort.

Knowing that PD remains a challenge in neurology due to the lack of an etiological treatment, as well as a therapy that significantly slows the progression of dopaminergic neurodegeneration, maintaining a decent QoL is important. This finding could be clinically significant if motivating or enjoyable music becomes part of both the rehabilitation approach and the patients’ lifestyle.

## Figures and Tables

**Table 1 brainsci-11-00569-t001:** Characteristics of the study population.

Basic Parameters	Study Group		Control Group	
Number	16		16	
Gender	Males 4	Females 12	Males 6	Females 10
Mean Age (years)	67.1 (±5.9)		65.6 (±5.5)	
Hoehn and Yahr stage	I—0 II—6 III—10		I—2 II—6 III—8	

**Table 2 brainsci-11-00569-t002:** Structure of the rehabilitation program sessions.

Physical therapy: I—5 min: breathing exercises and stretching movements, aiming to improve self-awareness as well as focus and concentration II—20 min: stationary cycling or treadmill walking of moderate intensity at 60–75% of heart rate (with individual adjustments and increases during the program) for cardiovascular training III—20 min: exercise regimen composed of muscle strengthening, range of trunk and limbs movement in different combinations, while slowly increasing the degree of difficulty every day IV—10 min: exercises for standing and walking balance training V—10 min: relaxation through breathing and stretching exercises Between sessions, patients had 10 min breaks for rest and socialization
Art therapy: 50 min (alternating daily: painting, drawing, crafting, handwriting, dancing)

**Table 3 brainsci-11-00569-t003:** Comparison of initial and final PDQ-39 scale values in the study group.

PDQ-39—Domains	Mean	Standard Deviation	Standard Error Mean	*p* Value
Global PDQ-39—initial	58.75	31.941	7.985	0.001
Global PDQ-39—final	49.63	29.929	7.482	
Mobility—initial	18.75	11.024	2.756	0.001
Mobility—final	17.00	10.328	2.582	
ADL—initial	10.38	5.954	1.488	0.001
ADL—final	8.13	5.018	1.255	
Emotional well-being—initial	10.63	5.608	1.402	0.001
Emotional well-being—final	8.00	5.391	1.348	
Stigma—initial	3.13	2.094	0.523	0.003
Stigma—final	2.50	2.130	0.530	
Social support -initial	1.75	1.438	0.359	0.003
Social support final	1.44	0.810	0.202	
Cognition—initial	3.63	2.187	0.547	0.04
Cognition—final	3.38	2.187	0.547	
Communication—initial	4.50	3.098	0.775	0.001
Communication—final	3.75	3.173	0.793	
Bodily discomfort—initial	6.00	2.633	0.658	0.001
Bodily discomfort—final	4.50	2.477	0.619	

ADL—activities of daily living.

**Table 4 brainsci-11-00569-t004:** Comparison of initial and final PDQ-39 scale values in the control group.

PDQ-39—Domains	Mean	Standard Deviation Mean	Standard Error	*p* Value
Global PDQ-39—initial	51.50	25.129	6.282	0.001
Global PDQ-39—final	46.63	24.177	6.044	
Mobility—initial	14.75	8.046	2.011	0.001
Mobility—final	13.63	7.606	1.901	
ADL—initial	9.88	4.425	1.106	0.25
ADL—final	8.88	4.425	1.106	
Emotional well-being—initial	11.88	5.691	1.423	0.001
Emotional well-being—final	10.00	5.610	1.402	
Stigma—initial	2.25	1.844	0.461	0.02
Stigma—final	1.88	1.610	0.400	
Social support—initial	1.38	1.258	0.315	1
Social support final	1.38	1.258	0.315	
Cognition—initial	3.38	2.579	0.645	1
Cognition—final	3.38	2.579	0.645	
Communication—initial	2.63	2.187	0.547	0.16
Communication—final	2.50	2.129	0.532	
Bodily discomfort—initial	5.38	3.538	0.884	0.001
Bodily discomfort—final	4.63	3.222	0.806	

ADL—activities of daily living.

**Table 5 brainsci-11-00569-t005:** Comparison of the median values of differences (initial value—final value) between the study group and the control group.

PDQ-39—Domains	Control Group	Study Group	*p* Value
Global PDQ-39	5 (4, 6)	9.5 (7.25, 11.5)	0.0001
Mobility	1 (1, 1.75)	2 (1, 2.75)	0.06
ADL	1 (1, 1)	2.5 (1.25, 3)	0.002
Emotional well-being	2 (2, 2)	3 (2, 3)	0.001
Stigma	0 (0, 0.75)	0 (0, 1)	0.6
Social support	0 (0, 0)	0.5 (0, 1)	0.02
Cognition	0 (0, 0)	0 (0, 0.75)	0.23
Communication	0 (0, 0)	1 (0.25, 1)	0.002
Bodily discomfort	1 (0.25, 1)	1.5 (1, 2)	0.002

ADL—activities of daily living.

## Data Availability

The data presented in this study are available on request from the corresponding author. The data are not publicly available due to privacy and ethical reasons.

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
