# Peer review of "Music as Add-On Therapy in the Rehabilitation Program of Parkinson’s Disease Patients—A Romanian Pilot Study"

_brainsci, 2021, doi:10.3390/brainsci11050569_

Round 1

Reviewer 1 Report

This study aimed to examine the effect of combining physical therapy with listening to Mozart music on self-reported quality of life in people with Parkinson’s disease, in comparison with physical therapy alone.  The study found subjective improvements across multiple domains of quality of life as measured by the Parkinson’s disease questionnaire (PDQ39), and greater improvements were reported in the group exposed to music alongside kinesiotherapy.

This is an interesting topic and the findings are intriguing, but there are several major issues with this study and how it is reported, as outlined below.

  1. The study did not use a case-control design and should not be reported as such. At some point in the paper the authors describe it as a “pilot study” and this would be a more appropriate term to use in the title.
  2. It is unclear why only the PDQ-39 was measured as an outcome. The Mozart effect is supposed to involve enhancing temporal-spatial performance, but the PDQ is an instrument to measure quality of life - it is not clear how the two relate to each other, and why no objective measures of temporal and spatial performance were included.
  3. On a related note, have previous studies used the PDQ as an outcome measure for RAS or music-based therapies, and if so what are the findings?
  4. The outcome was not measured immediately after the 2 weeks of physical therapy, but after a further 2 weeks in which the experimental group listened to Mozart music daily without physical therapy. It is therefore not clear whether the reported effect has anything to do with the physical component of the therapy or could have been achieved with listening alone. Since the PDQ asks patients to reflect on their quality of life over the past month, it might have been more useful to omit the two weeks of additional music therapy and reassess both groups after 2 weeks of no further input.
  5. The rationale for choice of statistical tests is not clear. Why were within-group differences analysed using parametric t-tests while between-groups were compared using Mann-Whitney tests? It would be better to use a within/between design - e.g., ANOVA with group and timepoint as factors. Additionally, it is not clear if the t-tests were adjusted for multiple comparisons.
  6. Given that the authors acknowledge in the introduction and discussion that the effectiveness of music-based therapy is likely to be influenced by preference and familiarity, it would be interesting to plot the individual data points, so that the reader can see how many patients benefited from the intervention. It would also have been informative to collect data on participants’ preferences and familiarity with the music.
  7. I do not believe that the authors can make any claims about the Mozart effect specifically from their findings, because (i) Mozart was not compared with any different type of music (ii) the Mozart effect refers to temporal-spatial performance, which was not assessed in this study.  The findings are potentially interesting but the paper (and title) should be re-framed in terms of effects of music rather than “the Mozart effect”.
  8. The conclusions do not directly follow from the present findings and should be adjusted accordingly.

Specific comments:

  1. Line 45 - it is not clear whether this reference showing effects on non-motor symptoms refers to rhythmic auditory stimulation or music therapy.
  2. Lines 51-52 - “conclusions of studies showing that listening to pleasurable music produces…” - a reference is needed for this claim.
  3. Lines 73-74 - the authors note that the Mozart effect is controversial but do not explain why. The meta-analysis they cite concludes there is no clear evidence for such an effect, but I am not familiar with subsequent findings that may have refuted this.
  4. Lines 74-76 - please provide some details of what types of effects were found in previous studies with neurological populations.
  5. Lines 76-77 - the authors note that there is little previous research on the Mozart effect in Parkinson’s. They cite one reference but do not explain the relevance of this previous work.
  6. Lines 77-79 - the aim of the study is described as being to assess the effect of Mozart music on quality of life in Parkinson’s. Since the original Mozart effect, as described earlier in the introduction, involves temporal-spatial performance, it should be explained why the researchers have instead decided to focus on quality of life as an outcome.
  7. Table 1 contains errors. Gender should be reported as male/female rather than man/women, and the Hoehn & Yahr stages in the study group are incorrect (stage II is listed twice instead of stage II and stage III).
  8. Line 101-102. There appears to be a missing section of text after “Hoehn-Yarh stages >III were…”.
  9. Line 103 - a reference is needed for the PDQ-39.
  10. Throughout the results - the PDQ-39 domain should be correctly named as “mobility” rather than “motility”.
  11. The first part of the discussion repeats information presented in the introduction and can be condensed.

Reviewer 2 Report

This study is interesting to practitioners who work with people with PD, and generally well written. There is, however, some important information missing that compromises the overall trustworthiness of the study, and I fear there is a risk that you overestimate your results. The so called “Mozart effect” only affected a temporo-spatial ability for 10 minutes in healthy children. There was no mentioning of quality of life in the original work by Rauscher et al (1993). This has since then been re-evaluated by Pietschnig et al in the well written meta-analysis (2010), which is also mentioned by you (ref. 25). In my opinion, it is highly likely that any music of their own choice would have led to the same improvements, and not only by Mozart. This should have been discussed more thoroughly. 

There are several methodological considerations and risk of bias to discuss: 

Selection bias: How did you arrive at 32 participants, did you perform a sample size prior to the study? What were the inclusion criteria (I believe the word “excluded” is missing from line 102)? How were the patients approached, how were they recruited and by whom? Did everyone who was approached accept to participate or were there patients who rejected? This information is important.

I also suggest that you put the information about ethical considerations after the recruitment and not in the end. What was the setting of the study? I believe that 2,5 hours daily over 14 days is not possible with community-dwelling patients, which compromises the generalizability.

Allocation bias: How was the allocation performed and by whom? Was this person blinded to groups or involved in the data analysis? Were the patients able to choose groups? Were the groups similar at baseline? The groups did not seem to be equal initially regarding PDQ-39, for example 58.75 vs. 51.50 on PDQ Global (Table 2 and 3).

Treatment bias: What is the hypothesis regarding improved quality of life by the music of Mozart? What is it about Mozart’s music that supposedly may have an effect on QoL, and why not any music of their liking? There is indeed no question about that rhythmical music does help people with PD through auditory cueing as clearly outlined in the introduction section. But there is less information about how it could help with QoL. I therefore suggest at least 1-2 sentences about your hypothesis as to possible effects on QoL as this comes as a surprise in the aim.

To use the music of Mozart is a nice addition, but I suggest that you use the expression “Mozart effect” with caution since there is very weak evidence for this according to you reference Pietschnig et al, 2010.

Was the intervention provided as a class or individually? If as a class, how many people were in each class? Was the program individualized? Was it progressed during the intervention period – and if so, how was this progression determined? The intervention lasted 2,5 hours every day but is very poorly described. I suggest you put details in a supplementary file as there is no registered study plan to be found.  How did you make sure that the patients listened to Mozart 2,5 hours daily at home?

Analytics bias: Regarding the statistical calculations, did you test the data for normality prior to the t-test? And if the data was not normally distributed, how did you handle this?

Measurement bias: The use of PDQ-39 to assess quality of life is relevant and the measurement is reliable and valid. It is, however, a very subjective measure and should be interpreted with caution. Are the results really clinically relevant or just statistically significant? 

All of these questions need answers to improve the credibility of your work. 

Round 2

Reviewer 1 Report

Please see my comments in the attached document. 

Reviewer 2 Report

Thank you for the revised manuscript. 

I think the revision has improved the credibility and the design is now correct (pilot study instead of case-control study). A minor remark: I think you should exchange the reference from Pohl et al from 2013 (a feasibility study with 12 participants) to the RCT from 2020 which in 46 patients confirmed the short-term effects on QoL (Group-based music intervention in Parkinson’s disease – findings from a mixed-methods study (sagepub.com)), and change line 254 to "duration of 4 to 12 weeks". 

The conclusion could be improved as it also includes art therapy, this has not been mentioned at all in the main article and raises more questions. 

The text also needs some text editing as there are several minor mistakes, you need to proof read the manuscript carefully. 
